# Bulgarian Medicinal Extracts as Natural Inhibitors with Antiviral and Antibacterial Activity

**DOI:** 10.3390/plants11131666

**Published:** 2022-06-23

**Authors:** Ivanka Nikolova, Tsvetelina Paunova-Krasteva, Zdravka Petrova, Petar Grozdanov, Nadya Nikolova, Georgi Tsonev, Alexandros Triantafyllidis, Stoyan Andreev, Madlena Trepechova, Viktoria Milkova, Neli Vilhelmova-Ilieva

**Affiliations:** 1Department of Virology, The Stephan Angeloff Institute of Microbiology, Bulgarian Academy of Sciences, 26 Georgi Bonchev, 1113 Sofia, Bulgaria; inikolova@microbio.bas.bg (I.N.); zdr.z1971@abv.bg (Z.P.); nadyanik@yahoo.com (N.N.); georgi.g.conev@gmail.com (G.T.); grebul31@gmail.com (A.T.); titi.andreev@gmail.com (S.A.); madi_trepechova@yahoo.com (M.T.); 2Department of General Microbiology, The Stephan Angeloff Institute of Microbiology, Bulgarian Academy of Sciences, 26 Georgi Bonchev, 1113 Sofia, Bulgaria; pauny@abv.bg; 3Institute of Morphology, Pathology and Anthropology with Museum, Bulgarian Academy of Sciences, 25 Georgi Bonchev, 1113 Sofia, Bulgaria; 4Laboratory Center Pasteur, The Stephan Angeloff Institute of Microbiology, Bulgarian Academy of Sciences, 26 Georgi Bonchev, 1113 Sofia, Bulgaria; grozdanov@microbio.bas.bg; 591 German Language High School of Sofia, 1000 Sofia, Bulgaria; 6Institute of Physical Chemistry, Bulgarian Academy of Sciences, 1113 Sofia, Bulgaria; viktoria.milkova44@gmail.com

**Keywords:** natural extracts, poliovirus-1, human adenovirus-5, herpes simplex virus-1, human coronavirus, virucidal activity, viral adsorption, antimicrobial activity

## Abstract

Background: Bulgaria is a country with a wide range of medicinal plants, with uses in traditional medicine dating back for centuries. Methods: Disc diffusion assay was used to evaluate the antimicrobial activity of the plant extracts. A cytopathic effect inhibition test was used for the assessment of the antiviral activity of the extracts. The virucidal activity of the extracts, their influence on the stage of viral adsorption, and their protective effect on uninfected cells were reported using the end-point dilution method, and Δlgs was determined as compared to the untreated controls. Results: The results of the study reveal that the antibacterial potential of *G. glabra* and *H. perforatum* extracts in Gram-positive bacteria is more effective than in Gram-negative bacteria. When applied during the replication of HSV-1 and HCov-OC-43, only some of the extracts showed weak activity, with SI between 2 to 8.5. Almost all tested extracts inhibited the extracellular virions of the studied enveloped viruses (HSV-1 and HCov-OC-43) to a greater extent than of the non-enveloped viruses (PV-1 and HAdV-5). They inhibited the stage of viral adsorption (HSV-1) in the host cell (MDBK) to varying degrees and showed a protective effect on healthy cells (MDBK) before they were subjected to viral invasion (HSV-1). Conclusion: The antipathogenic potential of extracts of *H. perforatum* and *G. glabra* suggests their effectiveness as antimicrobial agents. All 13 extracts of the Bulgarian medicinal plants studied can be used to reduce viral yield in a wide range of viral infections.

## 1. Introduction

The most common causes of human diseases are viral and bacterial infections. In practice, different chemotherapeutics are used to treat different viral infections, but their application is accompanied by many side effects, and there is continual development, in varying degrees, of resistant strains that are not susceptible to the therapies used [1].

Parallel to this key step-increased risk of antibiotic resistance, the alarming tendency of the appearance of clinically resistant isolates require the discovery of novel alternative ways to resolve the problem. It is well known that a minimum inhibitory concentration (MIC) and sub-MIC concentrations of antibiotics influence the physicochemical characteristics of bacterial cells, their functions, the expression of some virulence genes, the change of the morphology of individual cells, the reduction in the biofilm growth or formation, and interfere with quorum sensing mechanisms, motility, or pigment production [2]. Most of the infectious diseases are incurable using conventional antibiotic therapy due to the emergence of multidrug-resistant strains [3]. These strains cause approximately 50% of the worldwide nosocomial infections [4]. It is the growing incidence of multidrug-resistant bacteria that is driving the search for new powerful antimicrobial agents to fight them. The development of new antibiotics with new modes of action is very limited [5]. In the battle against bacterial resistance, a key role is played by natural products, in particular, plant substances. Over the time, plants have developed various defense mechanisms against bacterial infections, some of which consist of the synthesis of various compounds that inhibit the degree of bacterial pathogenicity.

Moreover, the application of natural products in the treatment of viral infections has some advantages over chemotherapeutics. Medicinal plant extracts are more easily absorbed by the body, due to their natural origin, and show fewer side effects [6]. The development of resistant strains to such antiviral agents is hampered by their complex chemical structures and often, by their multi-stage modes of action [7].

Of the approximately 250,000 flowering plant species described, between 50,000 and 70,000 are known to be used worldwide in traditional and modern medicine [8].

Bulgaria is a country with great potential in terms of herbal extraction, despite its small territory. The Bulgarian flora includes about 4300 plant species, over 500 of which are rare, or endemic to the country or the Balkan region. Of these, about 845 species are medicinal plants, most of which are wild [6,9].

There is preserved information about medicinal plants and the preparation of folk remedies from different historical periods. The use of natural remedies in Bulgarian traditional medicine dates back for centuries [10,11]. They present a wide range of species diversity of medicinal plants that can be used for various diseases and symptoms [12,13]. The current state of traditional knowledge in Bulgaria is of interest for national and global studies [14,15,16,17,18,19,20,21,22,23,24].

Most of the plant secondary metabolites (alkaloids, flavonoids, polyphenols, saponins, glycosides, tannins, anthraquinones, and sesquiterpenoids) have healing properties and therefore, are used in ethnomedicine [25,26]. The metabolic plant products have therapeutic activities, including anticancer, antioxidant, antidiabetic, immunosuppressive, antifungal, anti-inflammatory, antimalarial, anti-oomycete, antibacterial, anti-fever, anti-diabetic, insecticidal, anti-biofilm, and antiviral properties [21,26,27,28,29,30,31]. Their bioactive plant components can be used as a target for the identification of new chemical structures with antibacterial and antiviral activity [28,32].

Some flavonoids are able to inhibit virulence factors, as well as other forms of microbial threats, e.g., biofilm formation. In addition, some plant flavonoids have the ability to reverse antibiotic resistance and enhance the action of current antibiotic drugs [33]. The antimicrobial activity of polyphenols is associated with the inhibition of hydrolytic enzymes (proteases and carbohydrolases), as well as the inactivation of microbial adhesins, carbohydrates, cellular transport proteins in envelopes, etc. [34].

Medicinal plants are applied in different forms, depending on the symptoms—in the form of infusions, tinctures, syrups, dry extracts, oils, ointments and balms, or applied directly on the skin [35]. 

The aim of the present study was to investigate in vitro antibacterial and antiviral activity at different stages of viral infection in host cells of 13 crude extracts from medicinal plants.

## 2. Results

To validate some antimicrobial aspects of traditional uses of 13 Bulgarian plant extracts (Table 1), water–ethanol extracts were examined on Gram-negative and Gram-positive bacterial strains. One of the most popular antibacterial susceptibility tests, namely the agar disk-diffusion method, was performed to determine the activity of tested extracts. This method provides important information about the bacterial classification, depending on the sensitivity or resistance to the tested plant substances, based on the diameter of the inhibition zone [36]. 

Results of the study revealed that the antibacterial potential in Gram-positive bacteria was more efficient than in Gram-negative bacteria (Table 1, Figure 1).

**Table 1 plants-11-01666-t001:** Inhibition zone testing (mm) of the examined plant extracts against Gram-positive and Gram-negative bacterial strains.

Plant Extracts	Concentrations	Diameter of the Inhibition Zones (mm)				
		*B. subtilis*	*S. aureus*	*S. saprophyticus*	*E. coli* 25,922	*E. coli* 420
*Glycyrrhiza* *glabra* L.	10 mg/mL	-	-	-	-	-
	20 mg/mL	9 ± 0.7	-	-	-	-
	40 mg/mL	11 ± 0.6	12 ± 0.7	-	-	-
*Hypericum* *perforatum*	10 mg/mL	-	-	-	-	-
	20 mg/mL	9 ± 0.5		-	-	-
	40 mg/mL	10 ± 0	10 ± 0.5	-	-	-

The values of three parallel measurements are mean ± SD; -: not detected.

The absence of growth in *B. subtilis* and *S. aureus* around the discs is an indication of inhibition of the tested extracts (Figure 1). The anti-microbial inhibition was observed in the extracts from *H. perforatum*, Figure 1B,D, and *G. glabra*, Figure 1A,C, for two of the three tested concentrations. *H. perforatum* (9–10 mm) and *G. glabra* (9 mm) moderated anti-microbial activity (Table 1). Higher anti-microbial potential (11–12 mm) against both Gram-positive bacterial strains was demonstrated in *G. glabra*. In contrast, the plant extracts did not demonstrate inhibitory activity in Gram-negative bacteria, regardless of the tested concentrations. The antibiotic gentamicin was used as a positive control, which produced zones ranging from 25 to 30 mm, Figure 1B,D.

The cytotoxicity of the 13 extracts against 3 cell lines—HEp-2, MDBK, and HCT-8— was determined. Cytotoxic concentrations of 50% (CC_50_) and maximum tolerable concentrations (MTC) of the extracts were fixed. When comparing the cytotoxicity against the cell lines used, it was noticed that in all three cell types, the weakest cytotoxicity was shown by *A. officinalis* extract, with values for CC_50_ > 2000 µg/mL. For all extracts, the highest cytotoxicity was shown in *M. chamomilla* for all the three studied cell lines, being the most pronounced against HEp-2 cells. The maximum tolerated concentrations of the substances required for most of the following experiments were also determined (Table 2).

After determining the non-cytotoxic range of the 13 extracts, they were examined for their effect on the replication of 4 structurally different viral strains. As a model of a non- enveloped DNA virus, we used human adenovirus-5 (HAdV-5); as a DNA-enveloped virus, human herpes simplex virus type 1 (HSV-1); as an RNA-non-enveloped virus, poliovirus type 1 (PV-1); and as an RNA-enveloped virus, human coronavirus (HCov-OC43).

Applied during the replication of human herpes virus type 1, 6 of the extracts showed low activity. The strongest effect was shown by the extracts of garlic, with SI = 8.5, *A. hippocastani* (SI = 7.1), and *G. glabra* (SI = 6.8). The extracts of basil (SI = 4.5), chamomile (SI = 3.6), and thyme (SI = 2.3) showed lower activity.

Regarding the replication of HCov-OC43, only five extracts showed low activity; the most active extracts were *G. glabra*, with SI = 4.55, and *A. hippocastani* (SI = 3.73). Extracts of *A. sativum*, *P. reptans*, and *S. nigra* showed a lower selective index at 2. In the replication of the other viral strains tested, none of the extracts showed an effect on the intracellular replicative cycle (Table 3).

**Table 2 plants-11-01666-t002:** Cytotoxicity of extracts against HEp-2, MDBK, and HCT-8 cell lines.

Extract	Cytotoxicity (µg/mL)					
	HEp-2		MDBK		HCT-8	
	CC_50_ * Mean ± SD **	MTC ***	CC_50_ * Mean ± SD **	MTC ***	CC_50_ * Mean ± SD **	MTC ***
*A. hippocastani*	965.0 ± 21.7	400.0	1450.0 ± 33.4	1000.0	1420.0 ± 46.2	800.0
*G. glabra*	1280.0 ± 34.3	550.0	1850.0 ± 45.2	1000.0	1820.0 ± 24.5	1000.0
*O. basilicum*	1055.0 ± 16.6	550.0	1530.0 ± 29.7	1000.0	1300.0 ± 37.2	1000.0
*A. sativum*	1320.0 ± 22.5	550.0	2200.0 ± 38.5	2000.0	1880.0 ± 55.7	1200.0
*P. reptans*	1540.0 ± 36.2	1000.0	1890.0 ± 36.7	200.0	1880.0 ± 37.1	200.0
*M. chamomilla*	920.0± 18.5	500.0	1300.0 ± 41.4	1000.0	1150.0 ± 44.6	1000.0
*T. vulgaris*	1500.0 ± 21.8	1000.0	1500.0 ± 18.2	500.0	1100.0 ± 16.8	400.0
*A. officinalis*	2030.0 ± 34.2	1000.0	2450.0 ± 38.9	2000.0	2300.0 ± 62.4	1500.0
*R. canina*	1800.0 ± 26.3	1200.0	1700.0 ± 39.5	300.0	1600.0 ± 37.6	280.0
*A. annua*	1990.0 ± 41.2	1000.0	1320.0 ± 22.2	350.0	1200.0 ± 20.1	320.0
*S. nigra*	1440.0 ± 19.5	1000.0	2100.0 ± 47.1	1000.0	1900.0 ± 48.3	1000.0
*H. perforatum*	1650.0 ± 28.3	1000.0	1320.0 ± 32.4	1000.0	1220.0 ± 28.5	1000.0
*P. lanceolata*	1620.0 ± 23.5	1000.0	1980.0 ± 56.2	1000.0	1700.0 ± 46.2	1000.0

* CC_50_—cytotoxic concentrations 50%; ** SD—standard deviation; *** MTC—maximum tolerable concentration.

**Table 3 plants-11-01666-t003:** Antiviral activity against herpes simplex virus type 1 (Victoria strain) and human coronavirus (strain OC-43).

Extract	Antiviral Activity			
	HSV-1 (Victoria)		HCoV-OC43	
	IC_50_ * Mean ± SD ** (µg/mL)	SI ***	IC_50_ * Mean ± SD ** (µg/mL)	SI ***
*A. hippocastani*	205.0 ± 7.4	7.1	380.0 ± 9.5	3.7
*G. glabra*	272.0 ± 3.6	6.8	400.0 ± 12.5	4.5
*O. basilicum*	340.0 ± 5.2	4.5	-	-
*A. sativum*	260.0 ± 4.7	8.5	900.0 ± 18.5	2.1
*P. reptans*	-	-	890.0 ± 17.3	2.1
*M. chamomilla*	365.0 ± 6.2	3.6	-	-
*T. vulgaris*	650.0 ± 8.3	2.3	-	-
*S. nigra*	-	-	950.0 ± 32.7	2.0

* IC_50_—inhibitory concentration 50%; ** SD—standard deviation; *** SI—selectivity index.

After the results obtained for the replication cycle of viruses, the possibility of the ability of extracts to affect viral particles at other stages of their multiplication was investigated. The virucidal effect of the extracts on the extracellular virions of the four types of viruses included in the study was demonstrated.

Compared to the PV-1 virions, 4 extracts (*T. vulgaris*, *P. reptans*, *R. canina*, *S. nigra*) showed a weak inhibitory effect, with a decrease in the viral yield by about lg = 1, and with a longer incubation time (60–120 min) (Table 4).

Compared to human adenovirus virions, none of the studied extracts showed a significant inhibition at any of the observed time intervals (Table 5).

**Table 4 plants-11-01666-t004:** Virucidal activity of extracts against poliovirus type 1 (PV-1) virions.

Extract	Δlg			
	15 min	60 min	90 min	120 min
*A. hippocastani*	0	0.5	0.8	0.8
*G. glabra*	0	0.5	0.5	0.5
*O. basilicum*	0	0.5	0.8	0.8
*A. sativum*	0	0	0	0
*P. reptans*	0	1.0	1.0	1.0
*M. chamomilla*	0	0.5	0.5	0.5
*T. vulgaris*	0.5	1.0	1.2	1.2
*A. officinalis*	0	0	0	0
*R. canina*	0	1.0	1.0	1.0
*A. annua*	0	0	0	0
*S. nigra*	0.5	0.9	1.0	1.0
*H. perforatum*	0	0	0	0
*P. lanceolata*	0	0.5	0.8	0.8

**Table 5 plants-11-01666-t005:** Virucidal activity of extracts against human adenovirus virions.

Extract	Δlg			
	15 min	60 min	90 min	120 min
*A. hippocastani*	0	0.3	0.3	0.3
*G. glabra*	0	0.5	0.5	0.5
*O. basilicum*	0	0	0	0
*A. sativum*	0	0	0.5	0.5
*P. reptans*	0	0.3	0.3	0.3
*M. chamomilla*	0	0.5	0.5	0.5
*T. vulgaris*	0.5	0.5	0.7	0.7
*A. officinalis*	0	0	0	0.5
*R. canina*	0	0	0.3	0.5
*A. annua*	0	0	0	0
*S. nigra*	0	0	0	0
*H. perforatum*	0	0.5	0.5	0.5
*P. lanceolata*	0	0.5	0.5	0.7

Regarding the direct action of the 13 natural extracts studied on the extracellular virions of coated viruses, the influence is significantly more pronounced than that on the non-coated viruses.

Compared to the HSV-1 virions, 7 of the extracts showed virucidal activity in the first time interval studied (15 min). The most noticeable was the effect of the extracts of *A. sativum*, *O. basilicum*, and *P. reptans* (Δlg = 2.5), followed by the extracts of *R. canina*, *A. hippocastani*, *H. perforatum*, and *A. officinalis* (Δlg = 1.75). The exposure time of 30 min for *T. vulgaris* (Δlg = 1.75) also showed significant activity; the *M. chamomilla* and *G. glabra* extracts significantly reduced the viral titer by Δlg = 2.0 when exposed for 90 min.

The general dependency observed in the action of all extracts is that with increasing time exposure, the activity of the extracts also increases. At the last follow-up time interval (120 min), the extracts of *A. sativum*, *P. reptans* and *T. vulgaris* showed the most pronounced activity, with a decrease in the viral titer by Δlg = 3.5. Extracts of *A. hippocastani*, *H. perforatum*, *A. officinalis*, and *O. basilicum* (Δlg = 3.25), and extracts of *R. canina* and *A. annua* (Δlg = 3.0) also showed a strong inhibitory effect. The inhibitory effect of *M. chamomilla* extract was also significant (Δlg = 2.75). Even at 120 min of exposure, the activity of the tested extracts of *S. nigra*, *G. glabra*, and *P. lanceolata* was weak (Δlg = 1.5) (Table 6).

In the study of the direct interaction of the extracts on the extracellular virions of HCoV-OC-43 during the first studied time interval (15 min), only *P. reptans* extract showed a significant inhibitory effect, with a decrease in the viral titer of Δlg = 1.75, and this effect is maintained at a constant rate throughout the exposure. The extracts of *M. chamomilla*, *A. hippocastani* (Δlg = 1.75), *P. lanceolata*, *H. perforatum*, and *T. vulgaris* (Δlg = 2.0) showed significant virucidal activity at 60 min after exposure. At 90 min, the activity of *G. glabra* increased to Δlg = 1.75. The extracts that showed noticeable activity for up to 90 min, retained it at a constant rate for 120 min. Plant extracts of *S. nigra*, *A. sativum*, *R. canina*, *A. annua*, and *A. officinalis* demonstrated low inhibitory activity, even after 120 min of interaction time (Table 7).

From the experiments conducted so far, it was seen that, in general, the natural extracts studied do not have much effect on the replication cycle of the investigated viruses. However, most extracts have a significant inhibitory effect on the viability of the extracellular virions of the enveloped viruses examined.

In the following experiments, we decided to investigate the effect of the 13 extracts on the adsorption step of HSV-1 virions of sensitive MDBK cells. The inhibitory activity of *A. hippocastani* extract is extremely noticeable. At 15 min of exposure, it decreased the viral titer by Δlg = 5.0, and this inhibition increased and became complete at Δlg = 6.0 after 120 min of exposure.

**Table 7 plants-11-01666-t007:** Virucidal activity of extracts against human coronavirus virions.

Extract	Δlg				
	15 min	30 min	60 min	90 min	120 min
*A. hippocastani*	1.25	1.25	1.75	1.75	1.75
*G. glabra*	1.5	1.5	1.5	1.75	1.75
*O. basilicum*	1.0	1.0	1.0	2.0	2.0
*A. sativum*	0.5	1.5	1.5	1.5	1.5
*P. reptans*	1.75	1.75	1.75	1.75	1.75
*M. chamomilla*	1.0	1.0	1.75	1.75	1.75
*T. vulgaris*	0.5	1.0	2.0	2.0	2.0
*A. officinalis*	0.5	1.0	1.0	1.5	1.5
*R. canina*	1.25	1.5	1.5	1.5	1.5
*A. annua*	0.75	0.75	1.0	1.0	1.0
*S. nigra*	0.75	0.75	0.75	1.0	1.0
*H. perforatum*	1.0	1.0	2.0	2.0	2.0
*P. lanceolata*	0.5	1.0	2.0	2.0	2.0

The extracts of *H. perforatum* (Δlg = 3 = 0) and *O. basilicum* (Δlg = 2.5) also showed an inhibitory effect on the adsorption step of HSV-1 virions at an exposure time of 15 min. At 30 min of exposure, the effect of *G. glabra* and *T. vulgaris* extracts was significant (Δlg = 1.75). At the third follow-up time interval (45 min), three more extracts showed inhibition at the stage of viral adsorption—*A. sativum*, *A. officinalis* (Δlg = 2.0), and *A. annua* (Δlg = 1.75). At the last time interval, all tested extracts affected the viral adsorption of MDBK cells, to varying degrees. As mentioned previously, the effect of *A. hippocastani* extract is most pronounced (Δlg = 6.0), followed by the effect of *H. perforatum* and *O. basilicum* extracts (Table 8).

So far, we have found that the studied extracts show the strongest effect on extracellular HSV-1 virions and the stage of their adsorption in the host cell. In the next step of our study, we decided to test whether the extracts have a protective effect on the membranes of healthy cells, thus making them immune to subsequent viral infection. As an experimental model, we used MDBK cells and a subsequent herpes infection. Again, different time intervals for the ongoing impact were monitored. The extract of *A. hippocastani* showed the strongest protective effect. At 15 min of exposure, it lead to a decrease in the viral titers of subsequent herpes infection, with lg = 3.0. The protective effect at 15 min still showed the superiority of extract of *A. sativum* (Δlg = 2.0). Extracts of *M. chamomilla* (Δlg = 2 = 0), *P. reptans*, and *G. glabra* (Δlg = 1.75) also possess a protective effect after 30 min of treatment. Prolonged pre-treatment of the cells with the extracts reduced the viral yield of the subsequent infection. At 120 min of exposure, the protective effect of the extract of *A. hippocastani* (Δlg = 5.5) and *A. sativum* and *M. chamomilla* (Δlg = 3.0) was noticeable. Extracts of *P. reptans* (Δlg = 2.5), *P. lanceolata*, *G. glabra*, *S. nigra* (Δlg = 2.0), as well as extracts of *R. canina* and *H. perforatum* (Δlg = 1.75) also showed a significant protective effects. The protective activity of the extracts from *T. vulgaris* (Δlg = 1.5) and *O. basilicum* (Δlg = 1.33) was weak, while the effect of the extracts of *A annua* and *A. officinalis* was insignificant (Table 9).

## 3. Discussion

Medicinal plants are important sources of phytochemicals that have a beneficial therapeutic effect on human health. While modern therapeutic agents are generally ineffective and have a number of serious adverse effects, medicinal plants have been used in medicine since ancient times, and they are well known for their ability to promote wound healing and prevent infections, without serious side effects. Thus, herbal therapy can be an alternative or complementary strategy for the treatment of many health problems [35]. That is why more and more scientific evidence is currently being sought to prove the effectiveness of herbal medicines. Despite the many experiments conducted, the obtained scientifically based data are not enough to fully explain the mechanisms of their action.

The anti-microbial potential likely depends on the cell wall composition, especially on the thickness of peptidoglycan layers. Moreover, variations in cell membrane permeability indicate the interactions of plant components and cell membranes [36,37].

*Hypercum perforatum L.* (*Hypericaceae*), is one of the most famous medicinal plants in the world. Its popularity is due to its wide range of medicinal applications [38,39,40]. In the literature, some authors also demonstrate the antimicrobial activity of *H. perforatum* extracts against Gram-positive bacteria [41,42,43], which is confirmed by our data. 

The agar diffusion method using paper discs, as used in our study for the in vitro examination of the antibacterial activity of aqueous *G. glabra* extracts, showed activity against *S. aureus* and *B. subtilis* in a dose-related pattern. This same trend has been reported from root and leave extracts showing effectiveness against *C. albicans* and all examined Gram-positive bacteria [44,45]. It has been demonstrated that the highest sensitivity against *Staphylococcus aureus* was exhibiting by the methanolic extract of *G. glabra*, with a distinct inhibition zone [46]. In vitro antimicrobial activity of ethanolic extracts of *G. glabra* was screened against Gram-positive bacteria and Gram-negative bacteria [47]. Previously, the antibacterial potential of glabridin from *G. glabra* towards Gram-negative and Gram-positive bacteria was demonstrated by Gupta et al. The highest activity among all extracts tested against Gram-positive bacteria, as well as mycobacterial strains, was registered for glabridin [48]. 

The experiments conducted in a system (in the absence of the virus) showed that some of the extracts studied have a protective effect on MDBK cell membranes. The observed effect is most likely the result of the binding or structural modification due to the compounds included in the studied extracts, which have cell membrane surface structures that are essential for the recognition and attachment of the virus to the cell. The effect is, to some extent, dependent on the time of exposure and to the varying degrees to which individual extracts cause it to increase with time. It is possible that a similar effect is due to the inhibitory effect of the tested extracts on the adsorption step. In this case, the effect of the extracts is monitored in the presence of HSV-1 virions. Our results generally show that the effect on the adsorption step is slightly more pronounced than the protective effect on healthy cells. This also suggests some effect on the viral structures required for the virus to attach to the susceptible cell, which enhances the inhibitory effect we report. A research team has proven by electron microscopy that *S. nigra* extract compromises viral envelopes, suggesting that this is the likely mechanism of action in infectious bronchitis virus virions [49]. *S. nigra* extract blocks the specific viral glycoproteins of the influenza virus and thus disrupts the process of attachment of the virus to the cell [50]. The extract of *A. hippocastani* [51] has also shown an effect on extracellular virions, and numerous experiments have been performed with *A. sativum* extracts to demonstrate its virucidal activity, as well as its inhibition of the attachment and entry of human, animal, and plant viruses into host cells [52,53,54,55].The plant inhibitory effects on the viral absorption, especially for *H. perforatum* and *G. glabra,* possess a correlation with the antibacterial inhibitory activity, which demonstrated their successful antiviral and antibacterial applications.

## 4. Materials and Methods

### 4.1. Cells

Madin–Darby bovine kidney (MDBK) cells and human epithelial type 2 (HEp-2) cells originating from human laryngeal carcinoma were obtained from the National Bank for Industrial Microorganisms and Cell Cultures, Sofia. The cell lines were grown in DMEM medium containing 10% fetal bovine serum (Gibco^TM^, Waltam, MA, USA), supplemented with 10 mM HEPES buffer (Merck, Darmstadt, Germany) and antibiotics (penicillin 100 IU/mL, streptomycin 100 μg/mL) in a CO_2_ incubator (HERA cell 150, Heraeus, Hanan, Germany) at 37 °C/5% CO_2_. 

Human colon carcinoma (HCT-8) cells were purchased from the American Type Culture Collection (ATCC). Permanent HCT-8 [HRT-18] (ATCC-CCL-244, LGC Standards) were maintained at 37 °C and 5% CO_2_ using sterile RPMI 1640 (Roswell Park Memorial Institute Medium, ATCC-30-2001) supplemented with 0.3 g/L L-glutamine (Sigma-Aldrich, Darmstadt, Germany), 10% horse serum (ATCC-30-2021), 100 UI penicillin, and streptomycin 0.1 mg/mL (Sigma-Aldrich).

### 4.2. Viruses

Herpes simplex virus type 1, Victoria strain (HSV-1), was received from Prof. S. Dundarov, the National Center of Infectious and Parasitic Diseases, Sofia. The virus was replicated in a confluent monolayer of MDBK cells in a maintenance solution of Dulbecco’s modified Eagles’s medium (DMEM) Gibco BRL, Paisley, Scotland, UK, plus 0.5% fetal bovine serum Gibco BRL, Scotland, UK. The infectious titer of stock virus was 10^8.5^CCID_50_/mL.

Human coronavirus OC-43 (HCoV-OC43) (ATCC: VR-1558) strain was propagated in HCT-8 cells in a maintenance solution of RPMI 1640, supplemented with 2% horse serum, 100 U/mL penicillin, and 100 μg/mL streptomycin. A total of 5 days after infection, the cells were lysed by 2 freeze and thaw cycles, and the virus was titrated according to the Reed and Muench formula. The infectious titer of stock virus was 10^5.5^CCID_50_/mL.

Human adenovirus (HAdV-5) was taken from the collection of the Stephan Angeloff Institute of Microbiology, BAS (Sofia, Bulgaria). The working viral suspension was obtained by culturing in HEp-2 cell culture and supporting medium (DMEM with 0.5% fetal calf serum, 10 mM HEPES and antibiotics) at 37 °C and 5% CO_2_ for 48 h to 90–100% of progress toward the viral cytopathic effect (CPE), and subsequently frozen. The infectious titer was 10^7.2^ CCID_50_/mL.

Poliovirus 1 (LSc-2ab strain) (PV1) was received from the collection of the Stephan Angeloff Institute of Microbiology, BAS (Sofia, Bulgaria), grown in confluent monolayer of HEp2 cells in a maintenance solution DMEM (Gibco^TM^, Waltam, MA, USA), supplemented by 0.5% bovine fetal serum (Gibco^TM^, Waltam, MA, USA), 10 mmol HEPES buffer (AppliChem GmbH, Darmstadt, Germany), and antibiotics (penicillin, 100 U/mL, streptomycin, 100 mg/mL), and incubated at 37 °C and 5% CO_2_ for 48 h to 90–100% progress toward the viral cytopathic effect (CPE), and subsequently frozen. The infectious titer was 10^7.0^ CCID_50_/mL.

### 4.3. Bacterial Strains and Growth Conditions 

Several bacterial strains have been used in the present study. *Escherichia coli* 25,922, *Staphylococcus aureus* 29,213, and *Staphylococcus saprophyticus* 15,305 were purchased from the American type culture collection (ATCC). *Bacillus subtilis* 168 was obtained from the Culture Collection, Institute of Microbiology, Bulgarian Academy of Sciences, and *Pseudomonas aeruginosa* PAO1 was used from the International Reference Panel [56]. All bacterial strains were stored in 8% DMSO at (−80 °C). For screening experiments, all strains were routinely grown on Muller Hinton Broth (HiMedia, India) overnight at 37 °C and transferred on Mueller Hinton Agar (MHA) slants (HiMedia, Laboratories, Mumbai, India). Before each experiment, fresh broth cultures were prepared from the slants. Incubation was implemented at 37 °C for 18 h. 

### 4.4. Researched Plant Material

A total of 13 dry water-ethanol (15 °C) extracts of Bulgarian medicinal plants produced and provided by Extractpharma LTD—one of the leading companies in Bulgaria for production and trade in food supplements, herbal products, liquid and dry extracts of medicinal plants, Sofia, Bulgaria (Table 10). The method of extraction includes saturation of the drug and percolation by the method of reverse countercurrent at a temperature of 40 °C for 18 h.

#### Production of Dry Extract

The standardized herbal mixture (according to the recipe) is extracted in an extraction installation on the principle of countercurrent, by performing a classic solid–liquid extraction. The extractant is water–ethanol 15°. The extraction is carried out at atmospheric pressure and a temperature of 40 °C. The resulting liquid herbal extract is concentrated to a certain percentage of dry matter, depending on the type of herb, and dried in dryers—(1) a vacuum drying plant, and (2) a powder drying plant—to obtain a dry extract. The extracted herb is not reused.

**Table 10 plants-11-01666-t010:** Plant extracts.

Plant Species	Area of the Collected Material	Biological Activities	References
*Plantago lanceolata*(ribwort plantain)	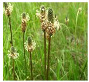	Stem	Used to treat wounds and inflammation; in diseases of the skin, respiratory organs, digestive system, reproduction, blood circulation; analgesic, antitumor, antioxidant, immunomodulatory activity.	[57,58]
*Matricaria**chamomilla* L.(chamomile)	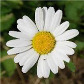	Flower	Antibacterial, antifungal, antioxidant activity; mosquito repellent and larvicidal effects.	[59,60]
*Glycyrrhiza**glabra* L.(licorice)	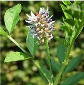	Root	Widely used for gastrointestinal problems (gastritis, peptic ulcers), cough, bronchitis, arthritis, respiratory infections, and tremors. Anti-inflammatory, antispasmodic, antioxidant, antimalarial, antifungal, antibacterial, antiviral, antidiabetic activity.	[61,62,63]
*Sambucus nigra*(elderberry)	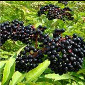	Fruit	Anti-inflammatory, antiviral, immunomodulatory activity.	[49,50]
*Potentilla**reptans*(creeping cinquefoil)	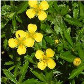	Stem	Antidiarrheal, anti-diabetic, antispasmodic, anti-inflammatory, antitumor, hepatoprotective, antioxidant, antifungal, antibacterial, antiviral activity.	[64]
*Ocimum**basilicum*(basil)	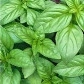	Stem	Anti-inflammatory, antioxidant, anthelmintic, antibacterial, antiviral activity.	[65,66,67]
*Thymus**vulgaris*(thyme)	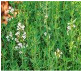	Stem	Anti-inflammatory, cardioprotective, neuroprotective, anti-osteoporotic, antispasmodic, anthelmintic, gastroprotective, antitumor, antioxidant, antibacterial, antiviral, immunoprotective activity.	[68,69,70,71]
*Artemisia**annua*(sweet wormwood)	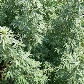	Stem	Anti-fever, antimalarial, antimicrobial, antiviral action.	[72,73]
*Rosa canina* L.(rosehip)	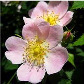	Fruit	Anit-osteoarthritis, anti-anxiety, anti-depression, analgesic, anti-diabetic, antihyperlipidemic, neuroprotective, anti-inflammatory, antioxidant, antitumor, antimicrobial effect.	[74,75]
*Allium sativum*(garlic)	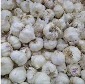	Bulb	In the prevention of infectious diseases, antiviral activity by inhibiting the entry of the virus, inhibition of viral RNA polymerase, reverse transcriptase, DNA synthesis; immunomodulatory activity.	[55]
*Althaea**officinalis*(white rose)	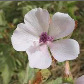	Root	Cough suppressing, anti-inflammatory, antioxidant, antimicrobial, immunomodulatory action.	[76,77]
*Aesculus**hippocastani*(horse chestnut)	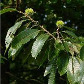	Seed	Vascular supporting, anti-inflammatory, antioxidant, immunomodulatory, virucidal, antiviral (against HSV-1, VSV, RSV, Dengue virus) activity.	[51,78]
*Hypericum**perforatum*(St. John’s wort)	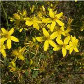	Stem	Burn healing, anti-anxiety, mild to moderate anti-depression, neuroprotection, anti-inflammatory, antioxidant, antiviral, antimicrobial, antitumoral, wound-healing and pain relief effects.	[79,80]

### 4.5. Disc Diffusion Assay

To evaluate the antimicrobial activity of the plant extracts, we used an established protocol, with some modifications [81]. Overnight bacterial suspensions were calibrated to 0.5 McFarland units, i.e., 1 × 10^8^ colony forming units per mL, using Densilameter II (Microlatest). Then 100 µL from the calibrated bacterial inoculums were streaked onto MHA plates. Plant extracts were dissolved in sterile deionized H_2_O at 3 different concentrations—10 mg/mL, 20 mg/mL, and 40 mg/mL. Sterile filter discs (Whatman paper, 6 mm/diameter) were impregnated with an appropriate concentration of the plant extracts and applied to the surface of MHA plates in duplicates. After incubation for 24 h at 37 °C, the zone of bacterial inhibition was measured by the appearance of a halo around the discs. The diameter of the inhibition zone (mm) was an indicator for the degree of antibacterial activity. As a positive control, separate discs containing the antibiotic gentamicin (15 µg/mL) were used.

### 4.6. Cytotoxicity Assay 

Confluent monolayer cell culture seeded in 96-well plates (Costar^®^, Corning Inc., Kennebunk, ME, USA) was treated with 0.1 mL/well supported medium containing decreasing concentrations of test extracts. The cells were incubated at 37 °C and 5% CO_2_ for 48 or 120 h. After microscopic evaluation, the medium containing the test compound was removed, and the cells were washed and incubated with neutral red at 37 °C for 3 h. After incubation, the neutral red dye was removed, and the cells were washed with PBS and 0.15 mL/well desorbing solution (1% glacial acetic acid and 49% ethanol in distilled water) was added. The optical density (OD) of each well was read at 540 nm in a microplate reader (Biotek Organon, West Chester, PA, USA). The 50% cytotoxic concentration (CC_50_) was defined as the concentration of the material that reduces the cell viability by 50% compared to the untreated controls. Each sample was tested in triplicate, with four wells for cell culture on a test sample.

The maximum tolerable concentration (MTC) of the extracts was determined by the concentration at which they do not affect the cell monolayer, and in the sample, it looks like the cells in the control sample (untreated with еxtracts).

### 4.7. Antiviral Activity Assay

A cytopathic effect (CPE) inhibition test was used for assessment of the antiviral activity of the tested compounds. A confluent cell monolayer in 96-well plates was infected with a 100 cell culture infectious dose of 50% (CCID_50_) in 0.1 mL (containing a different virus strain). After 60 or 120 min of virus adsorption, the non-adsorbed virus was removed and the tested compound was added in various concentrations; the cells were incubated for 48 h at 37 °C and 5% CO_2_ for PV-1, HAdV-5, and HSV-1, or 120 h at 33 °C and 5% CO_2_ for the HCov-OC-43 strain. The cytopathic effect was determined using a neutral red uptake assay, and the percentage of CPE inhibition for each concentration of the tested sample was calculated using the following formula:% CPE = [OD_test sample_ − OD_virus control_]/[OD_toxicity control_ − OD_virus control_] × 100,
where OD_test sample_ is the mean value of the ODs of the wells inoculated with the virus and treated with the test sample in the respective concentration, OD_virus control_ is the mean value of the ODs of the virus control wells (with no compound in the medium), and OD_toxicity control_ is the mean value of the ODs of the wells not inoculated with virus, but treated with the corresponding concentration of the test compound. The 50% inhibitory concentration (IC_50_) was defined as the concentration of the test substance that inhibited 50% of viral replication when compared to the virus control. The selectivity index (SI) was calculated from the ratio CC_50_/IC_50._

### 4.8. Virucidal Assay

Samples of 1 mL containing HSV-1 (10^5^ CCID_50_) and tested extract in its maximum tolerate concentration (MTC) were combined in a 1:1 ratio and subsequently stored at room temperature for different time intervals (15, 30, 60, 90, and 120 min). Then, the residual infectious virus content in each sample was determined by the end-point dilution method [82], and Δlgs compared to the untreated controls were evaluated.

### 4.9. Pre-Treatment of Healthy Cells

MDBK cell monolayers grown in 24-well cell culture plates (CELLSTAR, Greiner Bio-One) were treated for different time intervals (15, 30, 60, 90, and 120 min) at maximum tolerable concentration (MTC) of the tested extract in a maintenance medium (1 mL/well). After the above time intervals, the compounds were removed and the cells were washed with phosphate buffered saline (PBS) and inoculated with human herpesvirus type 1 (1000 CCID_50_ in 1 mL/well). After 120 min of adsorption, the non-adsorbed virus was removed, and the cells were coated with a maintenance medium. Samples were incubated at 37 °C and 5% CO_2_ for 24 h and, after freezing and thawing three times, infectious virus titers were determined by the final dilution method. Δlg was determined compared to the viral control (untreated with the compounds). Each sample was prepared in four replicates. 

### 4.10. Effect on Viral Adsorption

A group of 24-well plates containing MDBK monolayers were pre-cooled to 4 °C and inoculated with 10^5^ CCID_50_ of human herpesvirus type 1. In parallel, they were treated with tested extracts at their MTC and incubated at 4 °C for the time of virus adsorption. At various time intervals (15, 30, 45, and 60 min), the cells were washed with PBS to remove both the compound and the unattached virus; the cells were then covered with support medium and incubated at 37 °C and 5% CO_2_ for 24 h. After freezing and thawing three times, the infectious viral titer of each sample was determined by the final dilution method. Δlgs was determined compared to the viral control (untreated with the compounds). Each sample was prepared in four replicates.

### 4.11. Statistical Analysis

Data on cytotoxicity and antiviral effects were analyzed statistically. The values of CC_50_ and IC_50_ were presented as means ± standard deviation (SD). 

## 5. Conclusions

In the present study, our investigations demonstrated in vitro antimicrobial activity of 13 crude plant extracts. Plant samples were collected from different Bulgarian habitats. The aqueous extracts were evaluated for antimicrobial activities against five bacterial strains, some of them with important pathogenic potential, using disc diffusion assay. The antimicrobial method indicated that extracts from *H. perforatum* and *G. glabra* were more effective against Gram-positive bacteria than Gram-negative strains. 

Moreover, considerable evidence that Bulgarian *H. perforatum* and *G. glabra* extracts have antipathogenic potential reveal their effectiveness as antimicrobials with inhibitory potential.

All of our experiments show that the 13 extracts of medicinal Bulgarian plants studied by our team can be used to reduce the viral yield in a wide range of viral infections, especially by inhibiting extracellular virions, by inhibiting the stage of their attachment to the host cell, as well as to protect healthy cells before they have been subjected to viral invasion.

The identification, isolation, and structural characterization of the active components of these plants should be the objectives of further microbiological and virological investigations.

## Figures and Tables

**Figure 1 plants-11-01666-f001:**
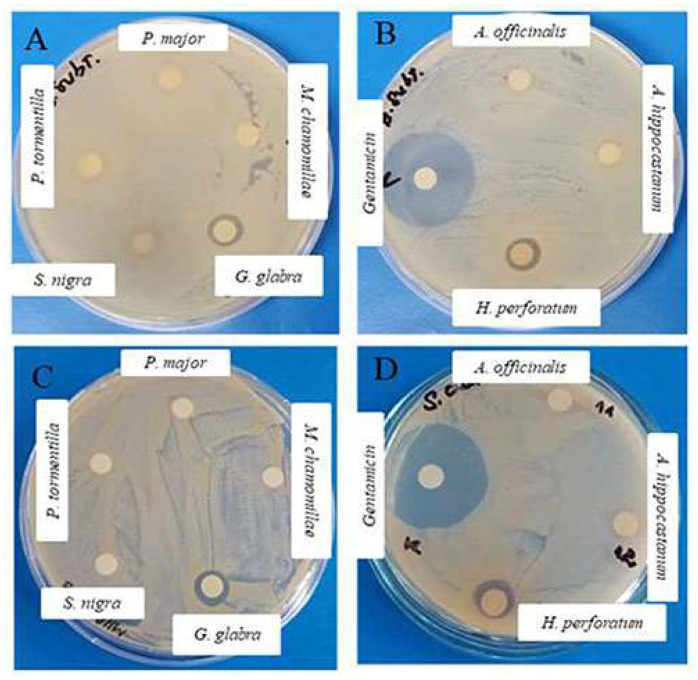
Determination of the anti-microbial activity of theextracts from *G. glabra* (**A**,**C**) and *H. perforatum* (**B**,**D**) against *Bacillus subtilis* 168 (**A**,**B**) and *Staphylococcus aureus* 29,213 (**C**,**D**). The results were obtained by the application of 40 mg/mL water–ethanol extracts using the disc diffusion method. Positive control: Gentamicin discs–15 µg/mL, (**B**,**D**). All experiments were repeated three times, and similar results were obtained.

**Table 6 plants-11-01666-t006:** Virucidal activity of extracts against human herpes simplex virus type 1 virions.

Extract	Δlg				
	15 min	30 min	60 min	90 min	120 min
*A. hippocastani*	1.75	2.0	2.75	3.25	3.25
*G. glabra*	0.25	0.5	1.25	1.5	1.5
*O. basilicum*	2.5	2.5	3.0	3.25	3.25
*A. sativum*	2.5	2.5	3.0	3.25	3.5
*P. reptans*	2.5	2.5	3.0	3.25	3.5
*M. chamomilla*	0.5	1.5	1.25	2.0	2.75
*T. vulgaris*	0.5	1.75	2.25	3.0	3.5
*A. officinalis*	1.75	2.0	2.5	3.25	3.25
*R. canina*	1.75	2.0	2.5	3.0	3.0
*A. annua*	0.5	0.5	0.5	2.0	3.0
*S. nigra*	0.5	1.0	1.5	1.5	1.5
*H. perforatum*	1.75	2.0	2.5	3.25	3.25
*P. lanceolata*	0.5	0.5	0.5	1.5	1.5

**Table 8 plants-11-01666-t008:** Influence of the extracts on the stage of adsorption of HSV-1 in sensitive MDBK cells.

Extract	Δlg			
	15 min	30 min	45 min	60 min
*A. hippocastani*	5.0	5.5	5.5	6.0
*G. glabra*	1.0	1.75	1.75	1.75
*O. basilicum*	2.5	2.5	3.0	3.5
*A. sativum*	1.25	1.25	2.0	2.25
*P. reptans*	2.0	2.75	2.75	2.75
*M. chamomilla*	2.0	2.0	2.0	2.0
*T. vulgaris*	1.5	1.75	1.75	1.75
*A. officinalis*	1.0	1.0	2.0	2.0
*R. canina*	1.0	1.0	1.5	2.0
*A. annua*	1.0	1.0	1.75	1.75
*S. nigra*	2.0	2.25	2.75	2.75
*H. perforatum*	3.0	3.5	3.5	3.5
*P. lanceolata*	1.0	1.0	1.75	1.75

**Table 9 plants-11-01666-t009:** Protective effect of pre-treatment of extracts on healthy MDBK cells and subsequent HSV-1 infection.

Extract	Δlg				
	15 min	30 min	60 min	90 min	120 min
*A. hippocastani*	3.0	3.75	4.25	5.0	5.5
*G. glabra*	1.5	1.75	2.0	2.0	2.0
*O. basilicum*	0.5	1.0	1.0	1.33	1.33
*A. sativum*	2.0	2.25	2.25	2.5	3.0
*P. reptans*	1.5	1.75	1.75	2.0	2.5
*M. chamomilla*	1.5	2.0	2.0	3.0	3.0
*T. vulgaris*	1.0	1.0	1.5	1.5	1.5
*A. officinalis*	0	0	0.5	0.5	0.5
*R. canina*	1.5	1.5	1.75	1.75	1.75
*A. annua*	1.0	1.0	1.0	1.0	1.0
*S. nigra*	1.0	1.0	1.5	2.0	2.0
*H. perforatum*	1.0	1.0	1.5	1.75	1.75
*P. lanceolata*	1.5	1.5	2.0	2.0	2.0

## Data Availability

Not applicable.

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
