# Peer review of "Bulgarian Medicinal Extracts as Natural Inhibitors with Antiviral and Antibacterial Activity"

_plants, 2022, doi:10.3390/plants11131666_

Round 1
Reviewer 1 Report
The article by Nikolova and collaborators is a study of the antibacterial and antiviral action of 13 extracts obtained from plants that are also found in the Bulgarian flora.
Although it is a well-conducted study in terms of testing antimicrobial, antiviral activity and also the cytotoxic effect, in order to be published I consider that it must be completed with the following aspects:
How were the plants collected and how were they identified? What parts of the plants were used?
What are the results of the chemical analysis of the extracts? What methods were used for these analyzes?
Are these extracts different in terms of chemical composition compared to similar extracts prepared from the same plants harvested from other areas?
What is the exact method of preparing the extracts?
What are the original elements of the research? Are these plants specific only to the Bulgarian flora? It is very important for the authors to emphasize these aspects of novelty because the literature is very rich in articles on research on the antibacterial and antiviral activity of the same plants from other regions.
Author Response
Point 1: How were the plants collected and how were they identified? What parts of the plants were used?
Response 1: Medicinal plants (herbs) are supplied by licensed herbalists. The plants are wild or cultivated. The requirements of the hazard analysis and critical control point (HACCP) system for control and qualification of herbs are observed. Table 10 shows which part of the plants was used to prepare the extracts.
Point 2: What are the results of the chemical analysis of the extracts? What methods were used for these analyzes?
Response 2: The results on the chemical analysis of the extracts are the aim of another of our studies. The data obtained have not yet been published and therefore cannot be cited in this manuscript. The experiments we conducted showed the presence of large amounts of polyphenols and flavonoids. The total flavonoid content (TFC) was determined using the aluminum chloride colorimetric method reported by Gouveia and Castilho (2011). The Folin – Ciocalteu assay was used to assess the total phenol content (TPC) of the extracts.
Gouveia, S.; Castilho, P.C. Characterisation of phenolic acid derivatives and flavonoids from different morphological parts of Helichrysum obconicum by a RP-HPLC–DAD-(−)–ESI-MSn method. Food Chemistry 2011, 129, 333-344.
McDonald, S.; Prenzler, P.D.; Antolovich, M.; Robards, K. Phenolic content and antioxidant
activity of olive extracts. Food chemistry 2001, 73, 73-84.
Point 3: Are these extracts different in terms of chemical composition compared to similar extracts prepared from the same plants harvested from other areas?
Response 3: The extracts studied by us do not differ significantly from similar extracts prepared from the same plants obtained from other regions. The content of the main ingredients is approximately the same, differing mainly in their percentage in the samples. In general, when collecting herbs from different geographical regions, there is always some difference in their composition. Even herbs collected from the same place, but in different years also give differences in their content, because the synthesis and accumulation of secondary metabolites in them depend on humidity, temperature, light and other environmental factors. I enclose several literature sources, proving as an example both the similar and partially different composition of extracts of Matricaria chamomilla L.
Singh, O., Khanam, Z., Misra, N., Srivastava, M.K. Chamomile (Matricaria chamomilla L.): An overview. Pharmacogn Rev. 2011, 5, 82-95.
Haghi, G., Hatami, A., Safaei, A., Mehran, M. Analysis of phenolic compounds in Matricaria chamomilla and its extracts by UPLC-UV. Res Pharm Sci. 2014, 9, 31-37.
Catani, M.V., Rinaldi, F., Tullio, V., Gasperi, V., Savini, I. Comparative analysis of phenolic composition of six commercially available Chamomile (Matricaria chamomilla L.) extracts: potential biological implication. Int J Mol Sci. 2021, 22, 10601.
Point 4: What is the exact method of
preparing the extracts?
Response 4: Production of dry extract: The standardized herbal mixture (according to the recipe) is extracted in an extraction installation on the principle of countercurrent, by performing a classic solid-liquid extraction. The extractant is water-ethanol 15 °. The extraction is carried out at atmospheric pressure and temperature 40 ° С. The resulting liquid herbal extract is concentrated to a certain percentage of dry matter, depending on the type of herb, and dried in dryers: 1) vacuum drying plant and 2) powder drying plant to obtain a dry extract. The extracted herb is not reused.
Point 5: What are the original elements of the research? Are these plants specific only to the Bulgarian flora? It is very important for the authors to emphasize these aspects of novelty because the literature is very rich in articles on research on the antibacterial and antiviral activity of the same plants from other regions.
Response 5: The studied extracts are from plants that have been used for centuries in Bulgarian traditional medicine. They are not endemic species and are distributed in other geographical areas. They are used to relieve various symptoms caused by bacterial, viral, fungal, parasitic diseases, as well as a general decrease in the activity of the immune system. The companies that provided us with the extracts - Extractpharma LTD and Mirta - Medicus LTD use them as ingredients in their products, which are sold commercially. The extracts studied by us are mainly in the composition of syrups stimulating the action of the immune system, syrups for children and adults against cough and affecting inflammation of the upper respiratory tract and tinctures with antiseptic action.
The fact that the extracts have a certain biological activity does not mean that they have a direct effect on the bacteria and viruses that cause the symptoms. The new contribution of our study is to monitor whether the extracts affect different stages of viral infection: the intracellular replicative cycle, extracellular virions, the stage of adsorption of viruses to cells and the protective effect of extracts on healthy cells before the presence of viral infection. For this purpose, we used four structural models of viruses (enveloped DNA, non-enveloped DNA, enveloped RNA and non-enveloped RNA). The results showed that the studied extracts did not significantly affect viral replication, but markedly inhibited (most likely non-specifically) extracellular virions, their attachment to the cell, and made the cell somewhat unsusceptible to viral infection. Such information about the action of the extracts is important when marketing medicinal products. It is important to know whether a drug is used for therapeutic or prophylactic purposes. Whether it acts directly on the cause or only on the symptoms of the disease.
Reviewer 2 Report
The manuscript entitled Bulgarian medicinal extracts as natural inhibitors with antiviral and antibacterial activity has many misleading results, missing the statistical approach in numerous tests and the novelty of the paper remains unclear. In my opinion this paper should be rejected. Some reasons for this statement are given below.
Lines 50-53 It is not mandatory that the mechanism of natural antimicrobial agents differs from the antibiotics as it is written in the introduction part.
Lines 66-69 appropriate reference for the given information is missing
Lines 94-99 complete paragraph is unnecessary in the Results part
Table 1- After the table it is noted that the experiments were performed in triplicate, but in the Materials and Methods it is mentioned that the experimental setup is conducted in duplicate.
Line 113 The strong anti-microbial inhibition was observed for H. perforatum??? The results of 9 and 10 mm are low to moderate activity, not strong.
Figure 1.- the quality of figure 1 is very low and the labeling is also unclear
Table 2. Missing the description of the abbevetrations after the table
In tables related to antiviral activity the results of statistical analysis are completely missing
The discussion part mainly refers to the antimicrobial activity and there is very poor discussion on the antiviral properties.
Generally, the novelty of the paper is not emphasized.
Author Response
Point 1: Lines 50-53 It is not mandatory that the mechanism of natural antimicrobial agents differs from the antibiotics as it is written in the introduction part.
Response 1: The mechanism of action of natural antimicrobials does not have to be different from that of existing antibiotics. But the development of new therapists with different mechanisms of action would expand the possibilities for therapy.
Point 2: Lines 66-69 appropriate reference for the given information is missing
Response 2: The reference for the information on the respective lines has already been presented.
Point 3: Lines 94-99 complete paragraph is unnecessary in the Results part
Response 3: The paragraph has been removed.
Point 4: Table 1- After the table it is noted that the experiments were performed in triplicate, but in the Materials and Methods it is mentioned that the experimental setup is conducted in duplicate.
Response 4: The experimental setup is conducted in triplicate.
Point 5: Line 113 The strong anti-microbial inhibition was observed for H. perforatum??? The results of 9 and 10 mm are low to moderate activity, not strong.
Response 5: The anti-microbial inhibition was observed in the extracts from H. perforatum, Figure 1 (B, D) and G. glabra, Figure 1 (A, C), for two of the three tested concentrations. H. perforatum (9 - 10 mm) and G. glabra (9 mm) moderated anti-microbial activity (Table 1). Higher anti-microbial potential (11-12 mm) against both Gram-positive bacterial strain was demonstrated in G. glabra.
Point 6: Figure 1.- the quality of figure 1 is very low and the labeling is also unclear
Response 6: The determination of the anti-microbial activity (Figure 1) was supported by addition of new figure with higher quality (600 dpi).
Point 7: Table 2. Missing the description of the abbevetrations after the table
Response 7: The abbreviations after Table 2 have been introduced.
Point 8: In tables related to antiviral activity the results of statistical analysis are completely missing
Response 8: Table 3, where the antiviral activity of the extracts is presented, includes statistical analysis of the results by presenting the standard deviation (SD), which is also indicated in the subsection Statistical analysis of Materials and methods. In other antiviral experiments (Tables 4 - 9), the change in viral titer was determined by the final dilution method of Reed and Muench.
Reed, L.J. and Muench, H. "A simple method of estimating fifty percent endpoints". The American Journal of Hygiene 1938. 27: 493–497.
Ramakrishnan, M.A. Determination of 50% endpoint titer using a simple formula. World J. Virol 2016, 5, 85-86. doi: 10.5501/wjv.v5.i2.85
Point 9: The discussion part mainly refers to the antimicrobial activity and there is very poor discussion on the antiviral properties.
Response 9: The results obtained by us related to the antiviral activity of the extracts were commented and compared with similar results obtained by other teams.
Point 10: Generally, the novelty of the paper is not emphasized.
Response 10: All the extracts we studied have proven different biological activities (Table 10). For some of them, antiviral properties have also been reported, but without tracing which stage of viral reproduction they affect. As stated in the manuscript, these extracts are part of a wide range of products of Extractpharma LTD and Mirta - Medicus LTD, which uses them as nutritional supplements, to treat various symptoms such as coughs, inflammation, to boost the immune system and etc. In these cases, conditions caused by bacteria, virus, fungal or parasitic disease are affected. The aim of our research is primarily to trace the effect of the extracts on the four structural models of viruses (enveloped DNA, non-enveloped DNA, enveloped RNA and non-enveloped RNA). The effect on different stages of virus infection was also studied: the intracellular replicative cycle, extracellular virions, the stage of adsorption of viruses to cells and the protective effect of extracts on healthy cells before the presence of viral infection.
Not every biological activity that is important in the treatment of an infectious disease means that there is antiviral activity.
For example, some time ago we studied Mursal tea extract, which is well known to be used for viral and bacterial infections accompanied by whooping cough due to its expectorant properties. In our antiviral experiments, however, the extract showed no activity against any of the viruses studied.
Therefore, the demonstration of antiviral activity at a certain stage of viral replication is important for the more correct application of natural therapists.
Reviewer 3 Report
After reviewing the paper entitled "Bulgarian medicinal extracts as natural inhibitors with antiviral and antibacterial activity", I can suggest some suggestions for the improvement of the whole paper.
The article is relevant and timely. The paper is straightforward, well-written and well structured. I find particularly interesting the discussion of results and conclusions even if at some points they are not fully supported by the results.
These are my suggestions for the Authors:
- Introduction part: please, provide explanations of MIC and sub-MIC values, and the main differences between them.
- Introduction part: wider paragraph of microbicide effect of plant secondary metabolites ic required;
- The aim of the study can be better explained; please, rewrite it;
- Results: line 101: Table 10 or Table 1?
- Table 1: indicate the interpretation of the results depending on inhibition zones (sensitivity, resistance, etc.)
- indicate why did you use two strains of E. coli, while other tested microorganisms only one strain? Indicate significance of the selected strains
- Table 2, Table 3 - indicate statistical analysis of the obtained results (mean, SD, etc.)
- line 343 rewrite reference "Balouiri et. al. (2016)" into number
- Disc diffusion assay:
Are you sure that you incubated the Bacillus strain at 37 ° C, for 24h? Bacillus subtilis and B. cereus are usually incubated on 30 °C?
What was a negative control?
Also, indicate the evaluation system for the obtained inhibition zone?
Author Response
Point 1: Introduction part: please, provide explanations of MIC and sub-MIC values, and the main differences between them.
Response 1: In the different environments, the contaminated surfaces are characterized by the presence of subtherapeutic concentrations of antibiotics. Thus, bacteria present in the environmental conditions are often subjected to drug amounts lower than the minimal inhibitory concentrations (MICs) (Andersson D and Hughes D. Nat. Rev. Microbiol. 2014;12:465–478). When bacterial cells grow in the presence of sub‐MICs, the antibiotics modify their physicochemical characteristics, their functions, and the expression of some virulence genes (Pompilio A, et. al., J. Med. Microbiol. 2010;59:76–81) sub‐MICs generally do not interfere with bacterial growth dynamics and microorganisms are exposed to stress.
Point 2: Introduction part: wider paragraph of microbicide effect of plant secondary metabolites ic required;
Response 2: The main mechanisms of antimicrobial action of secondary metabolites, which are in the highest concentration in plant extracts, are indicated.
Point 3: The aim of the study can be better explained; please, rewrite it;
Response 3: The aim of the manuscript has been rewritten.
Point 4: Results: line 101: Table 10 or Table 1?
Response 4: The results are presented in Table 1. The correction is made in the text.
Point 5: Table 1: indicate the interpretation of the results depending on inhibition zones (sensitivity, resistance, etc.)
Response 5: In accordance with this interpretation we checked the published CLSI guidelines, where it can be determined the susceptibility or resistance of the organism to each drug tested using different charts for different organisms, depending of zone size as susceptible (S), intermediate (I), or resistant (R) based on the interpretation chart only of applicable antibiotics or drugs (Clinical Laboratory Standards Institute. 2006. Performance standards for antimicrobial disk susceptibility tests; Approved standard—9th ed. CLSI document M2-A9. 26:1. Clinical Laboratory Standards Institute, Wayne, PA).
Point 6: indicate why did you use two strains of E. coli, while other tested microorganisms only one strain? Indicate significance of the selected strains
Response 6: The anti-microbial potential was investigated against model Gram negative strains (members of the family Enterobacteriaceae) Escherichia coli 420 and Escherichia coli 25922 (second is selected from CLSI and EUCAST for antimicrobial susceptibility testing), which are able to cause damage to industrial manufacturing, agriculture and human health. Many researchers alarmed that E. coli has developed multidrug resistance to many available antibiotics (Edeoga et. al., 2005; 4 (7):685–8; Pacheco et. al., A Search Antibact Agents. 2012) this motivate our findings to use more than one E. coli strains. In this regard, the current study did not demonstrate inhibitory activity in Gram-negative bacteria, regardless of the tested concentrations, where these findings agree with another reports about E. coli (Chuah E, et. al., J Microbiol. Res. 2014; 4(1):6–13; Dholaria M, et. al., J Emerg Technol Innov Res. 2018; 5(12):581–9).
Point 7: Table 2, Table 3 - indicate statistical analysis of the obtained results (mean, SD, etc.)
Response 7: Tables 2 and 3 have been corrected and the notations of the statistics used have been introduced.
Point 8: line 343 rewrite reference "Balouiri et. al. (2016)" into number
Response 8: The sentence has been rewritten.
Point 9: Disc diffusion assay:
Are you sure that you incubated the Bacillus strain at 37 ° C, for 24h? Bacillus subtilis and B. cereus are usually incubated on 30 °C?
Response 9: The strain B. subtilis 168 were purchased from National Bank of Industrial Microorganisms and Cell Cultures, Sofia, Bulgaria, which recommended growth at 37°C (Sotirova et. al., Curr Microbiol. 2012 Nov;65(5):534-41; Teper et. al., Materials (Basel). 2020;13(13):3037.).
Point 10: What was a negative control?
Response 10: As a negative control were prepared sterile discs impregnated with sterile deionized H2O.
Point 11: Also, indicate the evaluation system for the obtained inhibition zone?
Response 11: Same as comments in (Table 1: indicate the interpretation of the results depending on inhibition zones (sensitivity, resistance, etc.)
Round 2
Reviewer 1 Report
I consider that the authors improve the quality of the article.
Reviewer 2 Report
Dear Editor,
thank you once more for the opportunity to review the paper entitled "Bulgarian medicinal extracts as natural inhibitors with antiviral and antibacterial activity". I am positively surprised how the authors improved the quality of the paper. Now, I gladly recommend the publication of this paper in the present form.
Reviewer 3 Report
/